# Left Ventricle Contouring in Cardiac Images Based on Deep Reinforcement Learning

**Sixing Yin**                                                    YINSIXING@BUPT.EDU.CN

*Beijing Key Laboratory of Network System Architecture and Convergence, Beijing University of Posts and Telecommunications, Beijing 100876, China*

**Yameng Han**                                                   HAN_1997@BUPT.EDU.CN

*Beijing Key Laboratory of Network System Architecture and Convergence, Beijing University of Posts and Telecommunications, Beijing 100876, China*

**Judong Pan**                                                   JUDONG.PAN@UCSF.EDU

*University of California San Francisco, San Francisco, CA 94117, The United States*

**Yining Wang**                                                  YININGPUMC@163.COM

*State Key Laboratory of Complex Severe and Rare Diseases, Chinese Academy of Medical Sciences and Peking Union Medical College, Beijing 100730, China*

**Shufang Li**                                                    LISF@BUPT.EDU.CN

*Beijing Key Laboratory of Network System Architecture and Convergence, Beijing University of Posts and Telecommunications, Beijing 100876, China*

## Abstract

Assessment of the left ventricle segmentation in cardiac magnetic resonance imaging (MRI) is of crucial importance for cardiac disease diagnosis. However, conventional manual segmentation is a tedious task that requires excessive human effort, which makes automated segmentation highly desirable in practice to facilitate the process of clinical diagnosis. In this paper, we propose a novel reinforcement-learning-based framework for left ventricle contouring, which mimics how a cardiologist outlines the left ventricle along a specific trajectory in a cardiac image. Following the algorithm of proximal policy optimization (PPO), we train a policy network, which makes a stochastic decision on the agent's movement according to its local observation such that the generated trajectory matches the true contour of the left ventricle as much as possible. Moreover, we design a deep learning model with a customized loss function to generate the agent's landing spot (or coordinate of its initial position on a cardiac image). The experiment results show that the coordinate of the generated landing spot is sufficiently close to the true contour and the proposed reinforcement-learning-based approach outperforms the existing U-net model and its improved version, even with limited training set.

**Keywords:** Left ventricle contouring, cardiac image segmentation, deep reinforcement learning

## 1. Introduction

Deep learning has recently received extensive attention in a great number of areas (LeCun et al., 2015). In particular, tremendous effort has been made to automate medical imaging analysis and facilitate diagnosis through deep learning (Ravì et al., 2017). Most of the existing deep-learning models for image segmentation originates from CNNs (Krizhevsky

et al.). In 2015, Long et al. proposed the fully cnvolutional networks (FCNs) (Long et al., 2015). However, the FCN-based models could produce relatively sparse features and ignore global context. In response, encoder-decoder model is proposed for image segmentation (Badrinarayanan et al., 2017). U-net (Ronneberger et al., 2015) and V-net (Milletari et al., 2016) are two popular computational structures that are inspired by FCNs and the encoder-decoder model and there have been a great number of variants (Çiçek et al., 2016) (Zhou et al., 2018) (Li et al., 2018). To integrate more context, probabilistic graphical models such as conditional random fields are incorporated into CNNs and relevant models and algorithms have achieved encouraging results (Zheng et al., 2015) (Lin et al., 2016). Moreover, specialized information is found helpful for feature learning in medical image segmentation, such as anatomical positions (Bai et al., 2019).

However, due to the inevitable challenges of medical images, such as poor image quality, different imaging and segmentation protocols, and random variation among patients, such deep-learning-based approaches have still been considered untrustworthy and thus not yet been extensively put into clinical use. Such untrustworthiness comes mainly from the fact that the hidden features extracted by those end-to-end models based on deep learning fall short of human-level understandability and explainability, which inevitably leads to potential high risk in clinical trials. In this sense, a model that works in accordance with how human experts segment medical images is more preferable in practice. A human expert basically focuses on contouring an object in a medical image, which is also the routine work of a specialist doctor to provide explicit clue of clinical diagnosis.

With such motivation, in this paper, we focus on cardiac image segmentation and propose a novel approach for left ventricle contouring based on reinforcement learning (Sutton and Barto, 2018), which mimics how a cardiologist outlines the left ventricle along a specific trajectory in a cardiac image. The overall framework of the proposed approach is shown in Figure 1, where a robot tries to make the best movement on a cardiac image according to its local observation such that the trajectory it walks along sufficiently matches the contour of the left ventricle. The proposed approach is different from most of the existing supervised-learning-based approaches, which take medical image segmentation as pixel-level classification and focus largely on design for the end-to-end computational model[1]. Moreover, in order for the practicality of the proposed approach, we design a deep learning model with a customized loss function to generate the agent's landing spot, i.e., a coordinate on the cardiac image, from which the robot makes its first movement. Through comprehensive experiments, we show that the coordinate of the generated landing spot is sufficiently close to the true contour and the proposed reinforcement-learning-based approach outperforms the existing U-net model and its improved version, even with limited training set.

## 2. Methodology

### 2.1. Problem Definition

Instead of supervised-learning-based image segmentation to highlight the left ventricle, we focus on human-like contouring that mimics how a cardiologist outlines the left ventricle in

---

1. There have been previous work that make use of reinforcement learning for medical image segmentation. One can refer to Section A for more detail.

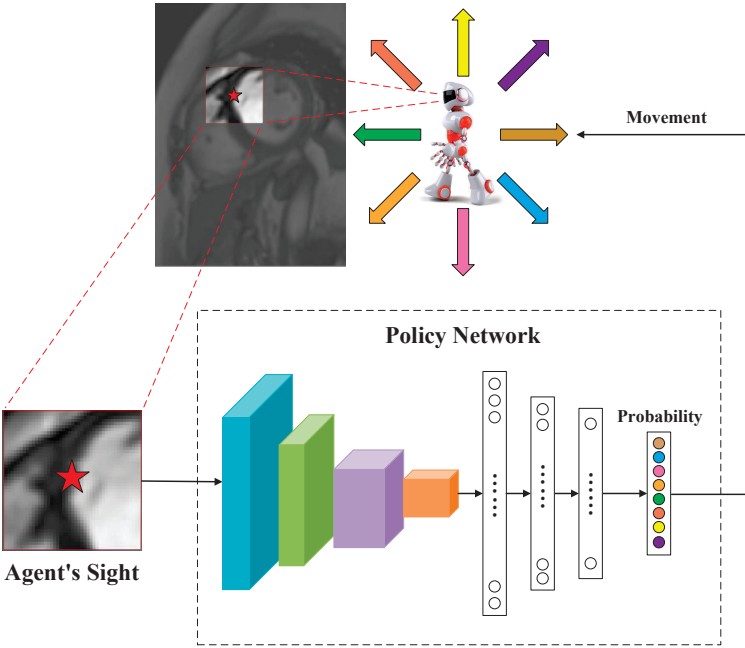

Figure 1: Overall framework of left ventricle contouring based on reinforcement learning.

a cardiac image. Since such a contour drawing process is simply moving a paintbrush, it can be analogized to a path finding problem, where a robot explores the unseen world and tries to find its way along the object boundary. In this problem, contouring the left ventricle along a specific trajectory in a cardiac image can be translated into a reinforcement learning problem, where an agent seeks to make the best decision according to its surrounding sight, i.e., a single-step movement, by interacting with an environment so as to travel as closely as possible to the true contour. Under the reinforcement learning framework, a policy network is trained to output the best movement according to the agent's local observation, as shown in Figure 1[2]. In the following part, the key elements of reinforcement learning framework for left ventricle contouring are formally defined.

### 2.1.1. ENVIRONMENT

An environment in left ventricle contouring can be defined as any cardiac image and its associated contour image. Here the whole cardiac image and the true contour of the left ventricle are unseen by the agent. Every time the agent interacts with such an environment by taking a single action, a reward is given by the environment as the feedback to indicate how good that action was, based on which the agent further updates its policy for a better decision, i.e., moving closer to the true contour.

---

2. In fact, the proposed method can be extended to multi-object segmentation by individually contouring each apart of the object in the training images. Hence, multiple single-object contouring tasks can be done in parallel.

### 2.1.2. State

In the reinforcement learning setting, a state is what the agent can observe, and based upon on which it can choose an appropriate action. In this paper, we specify that, being located at any pixel in a cardiac image, the agent can only myopically observe its surrounding sight rather than the whole image. We define the state $s$ as the $N \times N$ square patch centered by the pixel where agent is located. One advantage of such setting is that the policy network can be designed with a uniform structure since the input size is always $N \times N$.

### 2.1.3. Action

Since we analogize left ventricle contouring in a cardiac image to path finding in an unseen world, the agent's one-step movement can be defined as the action. Here we specify that at each step, the agent takes a one-pixel move towards one of its eight outbound direction and moves to one of its eight neighbouring pixels. Thus, the action $a$ with such definition is discrete and can be numbered with $a \in \{1, ..., 8\}$. Then with the agent's surrounding sight defined as state, Markovian property holds for state transition at each step, i.e., the next state depends only on the current one and the action taken by the agent.

### 2.1.4. Reward and Return

At each step, the environment gives a reward back to the agent based on its action. Since it is more desirable for the agent to stay as close as possible to the true contour while travelling, reward (denoted by $r_t$ for step $t$) can be defined as the negative distance between the agent's current position (denoted by $p_t$) and the corresponding position on the true contour (denoted by $p'_t$), i.e., $r_t = -|p_t - p'_t|$[3]. Therefore, the discounted accumulated return from step $t$ to the end of the episode (also termed "reward-to-go") can be defined as the discounted sum of negative distance over the entire episode, i.e., $R_t = -\sum_{\tau=t}^{T} \gamma^{\tau-t}|p_\tau - p'_\tau|$, where $\gamma \in (0, 1]$ and $T$ denote the discount factor and number of steps in one episode (which differs in different cardiac images), respectively.

### 2.1.5. Episode Termination

Normally, an episode can be terminated once the agent has finished the entire steps of actions, the total number of which is equal to the length of the true contour. However, an exception is that the agent might have moved illegitimately outside the cardiac image before the episode ends normally, which is a common case at the early stage of model training. In this case, we specify that the episode is terminated in advance and a reward of an extremely small negative value is given to the agent as a penalty for the illegal movement.

---

3. Apparently, with such definition of reward, the agent will select a movement that is closer to the true contour after being well trained. Here we use the "negative distance" as the reward because in a reinforcement learning setting, the agent's policy is optimized to maximize the total return. Moreover, positive reward definition could also be applicable if such distance mapped to a "score" by a decreasing function that always has positive values.

## 2.2. Policy Optimization

The previous definition for the elements of reinforcement learning implies a stochastic policy, i.e., the agent chooses its movement according to a probability distribution, which is output by the policy network with the agent's surrounding sight as input. We resort to proximal policy optimization (PPO) as the training algorithm: a policy-based algorithm which aims at iteratively improving the policy with a relatively small update (Schulman et al., 2017). With the PPO algorithm, the policy network is iteratively updated to maximize a clipped surrogate objective function defined as

$$L(\theta) = \mathbb{E}_{s,a\sim\pi_{\theta_k}}[\min(\frac{\pi_\theta(a|s)}{\pi_{\theta_k}(a|s)}A^{\pi_{\theta_k}}(s,a), \mathrm{CLIP}(\frac{\pi_\theta(a|s)}{\pi_{\theta_k}(a|s)}, 1-\epsilon, 1+\epsilon)A^{\pi_{\theta_k}}(s,a))] \quad (1)$$

for iteration $k$, where $\pi_\theta$ denote the policy under parameter $\theta$, $A^\pi(s,a)$ refers to the advantage of action $a$ for state $s$ under policy $\pi$, $\epsilon$ is a hyper-parameter that ranges from 0 to 1 to control the deviation from the current policy $\pi_{\theta_k}$ to the updated one $\pi_\theta$, and the clip function $\mathrm{CLIP}(x, b, u)$ limits $x$ within the range $[b, u]$. The PPO algorithm for left ventricle contouring is summarized in Algorithm 1, where $T_i$ refers to the total number of steps in episode trajectory $\tau_i$, i.e., the contour length. As training proceeds, parameters of both policy and value networks are iterative updated such that the agents turns smarter in locating the contour. One of the advantages with such a policy-based algorithm is that, even with limited training dataset, diversity of the collected episode trajectories can still be guaranteed by randomly choosing the agent's starting point on a contour in each iteration.

---

**Algorithm 1:** PPO for Left Ventricle Contouring

---

**Input:** Initial the parameters of policy network $\theta_0$ and value network $\phi_0$
**Output:** The parameters of policy network $\theta$ and the parameters of value network $\phi$
**for** $k = 0, 1, ....$ **do**

    For each training image, randomly select a position on its contour as the agent's starting point and run one episode by carrying out policy $\pi_{\theta_k}$. A set of episode trajectories $\mathcal{D}_k = \{\tau_i\}$ is then collected for the entire training images;

    Estimate advantage for each step based on the current value network:

$$A(s_t, a_t) = \sum_{t'=t}^{T} \gamma^{t'-t} r_{t'} - V_{\phi_k}(s_t);$$

    Update the parameters of the policy network to maximize the mean surrogate objective defined in (1): $\theta_{k+1} = \arg\max_\theta \frac{1}{|\mathcal{D}_k|} \sum_{\tau_i \in \mathcal{D}_k} \frac{1}{T_i} \sum_{t=0}^{T_i} L(\theta);$

    Update the parameters of the value network to minimize the mean squared error between the predicted value function and the estimated reward-to-go:

$$\phi_{k+1} = \arg\min_\phi \frac{1}{|\mathcal{D}_k|} \sum_{\tau_i \in \mathcal{D}_k} \frac{1}{T_i} \sum_{t=0}^{T_i} (V_{\phi_k}(s_t) - \sum_{t'=t}^{T} \gamma^{t'-t} r_{t'})^2.$$

**end**

---

## 2.3. Generating Landing Spot

During training, the agent's starting point is randomly chosen on a true contour. However, this is unrealistic for testing stage since the true contour of an unseen cardiac image is inaccessible. Therefore, where the agent should start to move on an unseen cardiac image still needs to be figured out. Apparently, the agent's landing spot should be as close to the true contour as possible. Otherwise, if the landing spot is excessively far away from the true contour, it is difficult for the agent to be back on track since its surrounding sight in this case could be significantly different from what it observed during training.

Intuitively, a deep learning model can be trained to generate the coordinate of the agent's appropriate landing spot. After inspecting all the cardiac images, we found that part of true contours always falls into the upper-right quadrant without exception. Therefore, we only focus on the upper-right sub-image with uniform size for each cardiac image to reduce the input size of the landing spot generator and improve training efficiency. The landing spot generator then takes a sub-image as input and is expected to output a coordinate that is sufficiently close to any point on the true contour. This can be done through training a supervised-learning model with the loss function defined as the mean minimum distance between the output coordinate and that of any point on the true contour, i.e., $L(\psi) = \frac{1}{B} \sum_{i=1}^{B} \min(d_1, ..., d_{T_i})$, where $d_t = |G_\psi(x) - p'_t|$ refers to the distance between the output coordinate and the $t$-th point $p'_t$ on the true contour of a sub-image, $G_\psi(x)$ refers to the landing spot generator with $\psi$ as parameters and sub-image $x$ as input, $T_i$ denotes the total length of the true contour that falls into the $i$-th sub-image and $B$ denotes the batch size.

The output coordinate of the landing spot generator is not necessarily an integer one, which is probably the case. Therefore, we simply round the output coordinate to its nearest integer in the sub-image. Then, the output coordinate still has to be translated back to that in the original (full-sized) cardiac image. This can be done by simply leaving the vertical coordinate unchanged while changing the horizontal coordinate $x$ following $x \Leftarrow x + (N_x - W)$, where $N_x$ refers to the horizontal size of the original cardiac image and $W$ denotes the width of the sub-image. According to the loss definition, the proposed landing spot generator also applies to datasets where left ventricles are located differently across cardiac images. In this case, landing spot generator can take the original full-size image as input and output the absolute coordinate of the landing spot. Thus, the coordinate translation will be unnecessary while the training process will be more time-consuming.

## 3. Experiment

### 3.1. Dataset

Our dataset is provided by the radiology department of a "triple-A-level" hospital in Beijing, China. It contains 100 two-dimensional MRI cardiac images in total, which was collected from 60 patients, and has non-uniform sizes of $208 \times 138$, $208 \times 150$, $208 \times 162$, $208 \times 168$, $208 \times 186$ and $208 \times 210$, respectively. Each cardiac image is associated with a contour image created by an experienced radiologists subspecialized in cardiovascular imaging as

Table 1: Contouring Performance Comparison

| Metrics | Average Dice Score | Average Hausdorff Distance |
|---|---|---|
| U-net without data augmentation | 0.896 (±0.075) | 14.158 (±13.386) |
| U-net with data augmentation | 0.935 (±0.052) | 11.966 (±11.784) |
| BL U-net without data augmentation | 0.899 (±0.075) | 9.098(±11.513) |
| BL U-net with data augmentation | 0.923 (±0.055) | 7.601 (±8.565) |
| Ours | 0.926 (±0.043) | 5.784 (±1.713) |

the ground truth. Throughout the experiment, 32 images are selected as the training set, 8 as the validation set while the remaining 60 as the testing set.

We consider each contour of the left ventricle as a sequential point set, which are counterclockwise sorted without loss of generality. In the original dataset, the contours marked by the cardiologists are commonly discontinuous "trajectories". In order to cater for the reinforcement learning setting, we have to first refine the original contours by interpolating sufficient extra points between two consecutive ones that are non-adjacent to each other to meet the continuity requirement, i.e., any point on a contour always have neighbours located within its $3 \times 3$ patch. Moreover, all the images are normalized with pixel values from 0 to 1 before being fed into the model.

### 3.2. Contouring Performance

With Dice score and Hausdorff distance as the two contouring performance metrics, we compare the proposed approach with two baselines: the original U-net model and an improved version with regional generalized Dice loss and boundary loss (termed BL U-net(Kervadec et al., 2019), both of which are intended for medical image segmentation based on supervised learning[4]. The result of contouring performance comparison with the same training set of 32 images is summarized in Table 1, where average Dice score and Hausdorff distance along with their standard deviation are presented. Moreover, data augmentation, which triples the train set to 96 images, is optionally conducted for U-net model training.

According to the numerical results in Table 1, the proposed approach for left ventricle contouring satisfactorily achieves 92.6% in average Dice score and 5.784 in average Hausdorff distance, even with a limited training set of only 32 images. Table 1 also shows that in general, the proposed approach outperforms both the U-net and BL U-net model with limited training set. An exception is that the proposed approach falls a little behind (while is still comparable to) the U-net model with data augmentation in average Dice score. Especially, for average Hausdorff distance, the proposed approach shows a significant advantage over the two baselines, even with data augmentation. This can be well explained by the rationale of the proposed approach, i.e., it focuses on drawing the contour of an object instead of making pixel-level classification.

The visual result for contouring performance evaluation with randomly selected testing images is shown in Figure 2, where the corresponding segmentation results are provided

---

4. Please refer Appendix B for implementation details.

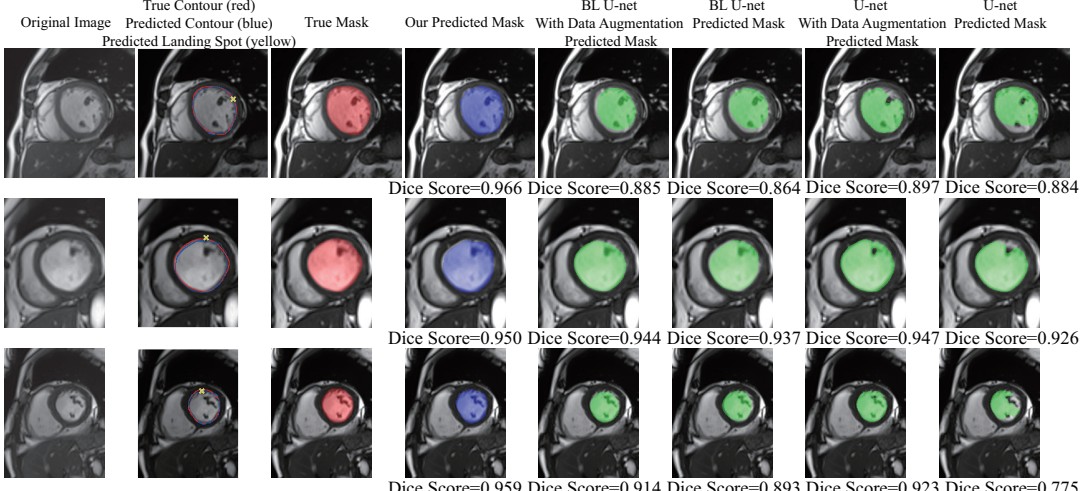

Figure 2: Visual comparison for segmentation performance with three sample images.

as well[5]. Although it is difficult to ensure exactly full consistency, the contours of the left ventricle generated via the proposed approach still maintain high resemblance to the ground-truth ones. It is also notable that even if the predicted landing spot deviates to a certain extent from the ground-truth contour (in the second row of Figure 2 as marked by the yellow cross), the agent is still able to get back on track so long as its sight covers part of the true contour. Moreover, the last two columns show that with limited training set, the segmentation performance of the original U-net model (even with data augmentation) is less satisfactory on cardiac images with irregular cavities on the left ventricles, where unexpected "grooves" and "holes" appear. This is because U-net might not work well in existence of fuzzy edges that make them non-differentiable. In contrast, left ventricles identified by the proposed approach are still highly consistent with the ground-truth ones. This is still owed to the fact that the proposed approach aims at drawing contours close to the ground-truth ones and the well-trained policy is barely distracted by irregular cavities.

## 4. Conclusion

In this paper, instead of end-to-end model design for left ventricle segmentation, we focus on imitating how a human expert outlines the left ventricle in a cardiac image. Following this notion, we think of left ventricle contouring as a path finding problem and propose a novel reinforcement-learning-based framework, where the agent interacts with the unknown environment and to make the movement based on its local observation such that its trajectory matches the true contour of the left ventricle as much as possible. We also design a deep learning model and customize the loss function to generate the agent's landing spot. The experiment results show that the coordinate of the generated landing spot is sufficiently close to the true contour and the proposed reinforcement-learning-based approach outperforms two U-net-based models, even with limited training set.

---

5. Since left ventricle contouring in cardiac images is analogized to a path finding problem, one can refer to (Han, 2021) for animated contouring results.

## Acknowledgments

The research was supported by Beijing Natural Science Foundation (Z210013).

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

## Appendix A. Reinforcement-learning-based Medical Image Segmentation

In fact, there have been several existing work that involves reinforcement learning in medical image segmentation (Zhou et al., 2021). However, in most of them, reinforcement learning is applied to either parameter search for segmentation network (Liao et al., 2020) or data augmentation (Qin et al., 2020). In (Mo et al., 2018), a policy model is trained to guide an agent to trace out the object boundary. However, it still works in a way of supervised learning. It is notable that one of the previous work is quite similar with the proposed method in this paper. In (Xiong et al., 2021), a reinforcement-learning model is proposed to imitate the human process of outlining left ventricle as well. Besides different alogirthms for model training, the main difference between (Xiong et al., 2021) and this paper lies in definition of state and reward. The state defined in (Xiong et al., 2021) contains the edge information in addition to the original grayscale image. The reward in (Xiong et al., 2021) is defined as the difference of Intersection over Union (IoU) in each step. To calculate such reward, a segmentation binary mask that comprises all the localized edge points must be maintained and updated in each step. In comparison, for our proposed method, state is simply defined as the square patch centered by the agent's position without extra information and reward is simply defined as location difference between the agent's position and the corresponding one on the true contour. Moreover, we also define episode termination for both training and testing stages and specify how to handle the early termination case.

## Appendix B. Implementation Details

Starting from the landing spot, the agent then sequentially makes its movement decision by following the policy network trained to draw the contour of the left ventricle. Throughout the experiments, the agent's eyesight, i.e., the input of the policy network, is fixed at a $21 \times 21$

square patch. The extremely small negative reward to penalize the agent for moving outside a cardiac image before the episode ends normally is set to $-400$[6]. We have $\gamma = 0.997$ for the discount factor. The whole model is trained with "Adam" as the optimizer and the initial learning rate of both policy and value networks is $3 \times 10^{-4}$, which exponentially decays as training proceeds. Moreover, the policy and value networks along with the landing spot generator share a similar structure, which contains two convolutional layers and three fully-connected layers. During model training, eight positions on the true contour are randomly selected as the agent's starting points for each cardiac image, which leads to a total of 256 experience trajectories in each iteration for the entire 32 training images. Moreover, during the testing stage, we specify that an episode is terminated if the agent have moved into the $3 \times 3$ square patch centered by its first five positions on the contour, which indicates that the agent has finished contouring and returned to the vicinity of the landing spot.

---

6. For the six different sizes of the cardiac images in our dataset ($208 \times 138$, $208 \times 150$, $208 \times 162$, $208 \times 168$, $208 \times 186$ and $208 \times 210$), such a penalty reward of 400 is much longer than the maximum diagonal distance of the cardiac images (less than 300 for the largest size $208 \times 210$), which is longer than the distance between any two points on a cardiac image. Thus, setting the penalty reward to $-400$ is able to sufficiently punish the agent's out-of-bound movement.

