# OpenReview forum: "Left Ventricle Contouring in Cardiac Images Based on Deep Reinforcement Learning"
_MIDL.io/2022/Conference — MIDL 2022_

### Official Review · Reviewer_Da5x · 2022-01-21

**Confidence:** 3
**Preliminary Rating:** 4
**Recommendation:** Poster

**Summary:**

The paper proposes a reinforcement learning based method for detecting left ventricle contours from cardiac MRI images. After estimating a start point, their method identifies other ones on the contour of the left ventricle. The paper reports better performance than U-Net despite being trained with less data.


**Strengths:**

(i)The paper is well written and easy to understand

(ii) Despite being trained with smaller size of dataset, their method outperforms U-Net in Dice score and HF distance

(iii) The proposed method mimics the behaviors of  human operators in labeling left ventricle contours, which improves trustworthiness of the method

**Weaknesses:**

(i) Performance comparison is limited with U-Net. More results should be reported from previous work, particularly from tracking based methods.
(ii) Recent literature on the use of reinforcement learning in left ventricle contour detection is not included. See [1].

[1] Xiong, Jingjing, et al. "Edge-Sensitive Left Ventricle Segmentation Using Deep Reinforcement Learning." Sensors 21.7 (2021): 2375.

**Deanonymize Review:**

no

**Detailed Comments:**

(i) Fig 3 can be improved by referring to methods on the top row.

(ii) Green regions in Fig 3 are not referred in its caption.

(iii) Does Table 1 shows merely the result of contour tracking or that of their overall method?

**Final Rating After The Rebuttal:**

5: Strong Accept

**Justification Of The Final Rating:**

The authors well addressed my concerns about the initial version of the paper, particularly adding performance comparison with previous work and revealing the difference between their method and the previous reinforcement leaning based method. I am happy with the current version of the paper.

**Paper Type:**

methodological development

**Questions To Address In The Rebuttal:**

(i) There is no performance evaluation given for the method selecting the start point of tracking. So, it is not clear if Table 1 reflects the performance of contour tracking or overall system. The authors should explain it.

**Special Issue:**

no

---

### Official Review · Reviewer_M8X2 · 2022-01-22

**Confidence:** 5
**Preliminary Rating:** 2
**Recommendation:** Poster

**Summary:**

The paper proposed to use a deep reinforcement-learning-based framework for cardiac magnetic resonance imaging (MRI). Proximal policy optimization
(PPO) is used for training. A network with a new loss is introduced to optimise the initial starting position of the agents. Comparisons are made to a conventional U-Net.

**Strengths:**

- the paper is reasonable well written and clearly presented
- the combination of known approaches is sound and the results show that it seems to work
- code seems to be available and reproducibility is high also because of publicly available MR data (though the used data is private)
- RL can achieve good results with limited data

**Weaknesses:**

- methodological insights are limited, not going far beyond an ordinary application of a moving agent to do iterative refinement
- I am not sure if really policies are optimised here or if the results are simply the result of iterative transformation steps
- Figure 2 does not add a lot to the paper
- the results are marginally better than using a simple U-Net
- Figure 3 should made clearer regarding what is the result of which method.

**Deanonymize Review:**

no

**Detailed Comments:**

please consider discussing these related works as well in the context of your approach:

Bai W, Chen C, Tarroni G, Duan J, Guitton F, Petersen SE, Guo Y, Matthews PM, Rueckert D. Self-supervised learning for cardiac mr image segmentation by anatomical position prediction. InInternational Conference on Medical Image Computing and Computer-Assisted Intervention 2019 Oct 13 (pp. 541-549). Springer, Cham.

Mo Y, Liu F, McIlwraith D, Yang G, Zhang J, He T, Guo Y. The deep poincaré map: A novel approach for left ventricle segmentation. InInternational Conference on Medical Image Computing and Computer-Assisted Intervention 2018 Sep 16 (pp. 561-568). Springer, Cham.

Xiong J, Po LM, Cheung KW, Xian P, Zhao Y, Rehman YA, Zhang Y. Edge-Sensitive Left Ventricle Segmentation Using Deep Reinforcement Learning. Sensors. 2021 Jan;21(7):2375.

**Final Rating After The Rebuttal:**

4: Weak Accept

**Justification Of The Final Rating:**

Thank you for addressing the points raised above and adding clarifications to the presented work. I have no further comments.
Thank you for addressing the points raised above and adding clarifications to the presented work. I have no further comments.

**Paper Type:**

methodological development

**Questions To Address In The Rebuttal:**

Please discuss how this approach would differ from a localisation network combined with a simple iterative transformer network. How are policies optimised here beyond stochastic search?

since result variances are provided, could you please comment how these come about and would it be possible to derive significance analysis from them?

**Special Issue:**

no

---

### Official Review · Reviewer_prpY · 2022-01-24

**Confidence:** 4
**Preliminary Rating:** 4
**Recommendation:** Oral, Poster

**Summary:**

The authors propose to use a reinforcement learning (RL) based approach to perform left ventricle segmentation in cardiac images. They do this via contouring, similar to how a clinical expert would annotate it. Firstly, the landing spot of the contour is predicted by a network learned to predict this. Next, the contouring of the object (left ventricle) from this predicted spot is done by leveraging an RL-based approach called proximal policy optimization (PPO), a policy gradient method. The contour is estimated using the policy predicted by a learned network. They evaluate this on a private cardiac MRI dataset and observe better performance than conventional end-to-end CNNs (U-Net) widely used for segmentation. Unlike CNNs, they observe this approach does not predict contours with unrealistic shapes.

**Strengths:**

1. It's an interesting idea to do segmentation via contouring with an RL-based approach.
2. Overall, it is a well-structured and written article. It is easy to follow.
3. The experiments and compared methods are alright, and the appropriate details are provided.
4. The visual results are well presented and demonstrate the inability of CNNs (U-Net) to figure out cavities and grooves in left ventricles and thereby predict erroneous segmentation masks. The RL-based approach does not make these mistakes and predicts more feasible contours close to the true contours (ground truths) than the end-to-end segmentation CNNs.

**Weaknesses:**

P1. In Figure 1, the RL framework presented depicts the policy network to output the probabilities for 8 possible discrete actions.
In methods, the authors mention the approach to be PPO, which predicts continuous action value. In PPO, the output is a mean value of the action modeled as a Gaussian distribution with fixed variance. The figure suggests it as a discrete action space model such as Deep Q-Network (DQN) or Double DQN (DDQN). In discrete models, we have a fixed set of possible actions, and the network outputs the corresponding Q values for all these actions, and we select the action that yields the highest Q value. While continuous action space models with PPO have two networks referred to as actor and critic networks which are not clear from Figure 1. Here, the actor-network outputs the mean value of Gaussian (action value) and critic network the advantage function value as referred to in Equation 1.

P2. The improvements in Dice are quite small. It would be interesting to evaluate this approach on harder to segment structures like the right ventricle, myocardium in cardiac images, or, on other difficult tasks like tumor segmentation in brain images.

P3. It is not clear how the absolute values for the negative rewards are determined. Can the authors please provide some intuition behind the selection of the values? Also, can they comment on if they use any positive rewards?

P4. How can this method be extended to multi-object segmentation? Can the authors please comment on it? How would one segment objects that require multiple contours, for instance, in cardiac images, the myocardium is segmented with 2 contours.

P5. The assumption that the landing spot will be on the top right corner of the image can be risky as this might change with each dataset. Can the authors please clarify this on how they ensure the top right patch contains the left ventricle?

P6. Currently, the comparison is limited. Some interesting comparisons could be: (a) articles that leverage the shape constraints in CNN training [1,2,3,4] to ensure the network predicts plausible segmentation masks and (b) works that perform interactive segmentation based on user hints [5,6].

[1] Oktay et al, “Anatomically constrained neural networks (ACNNs): application to cardiac image enhancement and segmentation”, IEEE TMI, 2017.

[2] Ravishankar et al, “Learning and incorporating shape models for semantic segmentation”, MICCAI, 2017.

[3] Dalca et al, “Anatomical priors in convolutional networks for unsupervised biomedical segmentation”, IEEE CVPR, 2018.

[4] Kervadec et al, “Boundary loss for highly unbalanced segmentation”, MIDL, 2019.

[5] Wang et al, "Deepigeos: a deep interactive geodesic framework for medical image segmentation", IEEE PAMI, 2018.

[6] Sakinis et al, "Interactive segmentation of medical images through fully convolutional neural networks", arxiv:1903.08205.


**Deanonymize Review:**

no

**Detailed Comments:**

Other minor comments are mentioned below:

1. What is the inference time of the proposed approach compared to U-Net.
2. Some of the PPO hyper parameters details are missing. It would be great if these details can be provided or the code can be made available.
3. One can use augmented images for training the proposed RL algorithm as well. It would be Interesting to see if it can provide some additional benefits.
4. The loss definition and training details for the landing spot and contour prediction are quite brief. It can be helpful to provide more details and intuition for the readers who want to re-implement it.
5. The authors could benefit from citing other RL based works used for medical image segmentation [7,8,9].

[7] Tiexin et al, “Automatic Data Augmentation via Deep Reinforcement Learning for Effective Kidney Tumor Segmentation”, IEEE ICASSP, 2020.

[8] Liao et al, “Iteratively-refined interactive 3d medical image segmentation with multi-agent reinforcement learning”, IEEE CVPR 2020.

[9] Zhou et al, “Deep reinforcement learning in medical imaging: A literature review”, Medical Image Analysis, 2021.

**Final Rating After The Rebuttal:**

5: Strong Accept

**Justification Of The Final Rating:**

The authors have addressed most of the concerns raised and have provided valid justifications/clarifications.
They have made the appropriate changes in the revised draft to include the suggested comments.
In future work, the authors can extend it for a journal for the problem of multi-object segmentation or harder problems like tumor/lesion segmentation.
I am happy with the revised draft.

**Paper Type:**

methodological development

**Questions To Address In The Rebuttal:**

It would be great if the authors can clarify or address the concerns stated in the weakness section namely the points [P1 to P6].
1. It is important to clarify the method as the figure does not exactly correspond to methods' description [P1].
2. It would be great if more comparisons in results could be provided [P6].
3. Other comments clarification would be nice as well [P2 to P5].

**Special Issue:**

yes

---

### Meta-Review · Area_Chair_LAD3 · 2022-02-16

**Recommendation:** Accept (Poster)
**Confidence:** 5

**Metareview:**

After addressing all reviewer comments appropriately, all reviewers recommend acceptance of the paper. The paper seems to be well written and the method and evaluation sound (in particular after addressing some of the reviewer's concerns).

I recommend to accept this paper.

---

### Decision · Program_Chairs · 2022-02-28

Accept